# Evaluation of Open-Source and Pre-Trained Deep Convolutional Neural Networks Suitable for Player Detection and Motion Analysis in Squash

**DOI:** 10.3390/s21134550

**Published:** 2021-07-02

**Authors:** Christopher Brumann, Markus Kukuk, Claus Reinsberger

**Affiliations:** 1Department of Computer Science, University of Applied Sciences and Arts Dortmund, 44139 Dortmund, Germany; markus.kukuk@fh-dortmund.de; 2Paderborn University, Department of Exercise and Health, Institute of Sports Medicine, 33098 Paderborn, Germany; reinsberger@sportmed.upb.de

**Keywords:** racket sports, sports analysis, video tracking, human pose estimation

## Abstract

In sport science, athlete tracking and motion analysis are essential for monitoring and optimizing training programs, with the goal of increasing success in competition and preventing injury. At present, contact-free, camera-based, multi-athlete detection and tracking have become a reality, mainly due to the advances in machine learning regarding computer vision and, specifically, advances in artificial convolutional neural networks (CNN), used for human pose estimation (HPE-CNN) in image sequences. Sport science in general, as well as coaches and athletes in particular, would greatly benefit from HPE-CNN-based tracking, but the sheer amount of HPE-CNNs available, as well as their complexity, pose a hurdle to the adoption of this new technology. It is unclear how many HPE-CNNs which are available at present are ready to use in out-of-the-box inference to squash, to what extent they allow motion analysis and if detections can easily be used to provide insight to coaches and athletes. Therefore, we conducted a systematic investigation of more than 250 HPE-CNNs. After applying our selection criteria of open-source, pre-trained, state-of-the-art and ready-to-use, five variants of three HPE-CNNs remained, and were evaluated in the context of motion analysis for the racket sport of squash. Specifically, we are interested in detecting player’s feet in videos from a single camera and investigated the detection accuracy of all HPE-CNNs. To that end, we created a ground-truth dataset from publicly available squash videos by developing our own annotation tool and manually labeling frames and events. We present heatmaps, which depict the court floor using a color scale and highlight areas according to the relative time for which a player occupied that location during matchplay. These are used to provide insight into detections. Finally, we created a decision flow chart to help sport scientists, coaches and athletes to decide which HPE-CNN is best for player detection and tracking in a given application scenario.

## 1. Introduction

Training is an integral part of sports. Well-planned and conscientiously executed adaptation mechanisms can lead to improvements in athletes’ performance, and an optimized training program can ultimately lead to more success in competition, while decreasing the risk of injury [1]. At present, training quality and effectiveness can be quantitatively evaluated and measured using different types of sensors. For physiological core measures, such as fitness and endurance wearable sensors, monitors for heart rate, blood pressure and oxygen level are available [2]. Likewise, for training aspects generally concerning movement and game tactics, motion sensors are available to measure velocity, acceleration and motion trajectories [3]. A classic example can be found in football (soccer), where team performance and collaboration is paramount and, therefore, individual player on-field locations, moves and motion paths are analyzed [4,5].

The SAGIT/Squash, introduced by Pers in 2008 and improved and used by Vučković et al. [6], represents an early example of camera-based player-tracking in squash. The system requires a downward-facing camera mounted on the ceiling, centered above the court. The SAGIT/Squash system was used for several studies [7,8,9].

As well as the classic tracking approaches evaluated by van der Kruk and Reijne [3], the development of convolutional neural networks (CNN) in the field of deep machine learning for computer vision may offer new approaches for detection and tracking applications in sport sciences. Classic applications for CNNs are for example recognizing and classifying images of handwritten digits [10]. Recently, CNNs found wide application in the field of medical imaging for applications such classification or segmentation [11]. In addition, deep learning has also been successfully applied in the area network data transmission and traffic classification technology [12,13]. For handball [14] and football [15] specifically, CNNs have been used for player detection and tracking.

Due to the increased interest in machine learning, the sheer number of human pose estimation convolutional neural networks (HPE-CNNs) available and their complexity pose a hurdle to the adoption and implementation of this new technology. It is unclear how many are open-source, pre-trained and, out-of-the-box, ready-to-use for player detection and motion analysis in squash. Therefore, we address the following research questions:RQ1: How many HPE-CNNs available today are ready to use for out-of-the-box inference on squash data for motion analysis?RQ2: To what extent and with what accuracy do they allow motion analysis in squash?RQ3: Can the data obtained from the HPE-CNNs selected in RQ2 be easily used to provide insight to coaches and athletes?

To answer the first question, we conducted a systematic investigation of more than 250 HPE-CNNs. After applying our selection criteria of open-source, pre-trained, state-of-the-art and ready-to-use HPE-CNNs, three of five variants remained. Regarding question two, we evaluate the detection accuracy of different HPE-CNNs. To evaluate and compare them, a labeled dataset is required, since no dedicated squash dataset is available. Therefore, we created a novel ground truth dataset for evaluation by developing and implementing a labeling tool. We accessed publicly available squash matches with different court conditions and annotated the athletes’ feet. For the last research question, we present heatmaps that reflect players’ motion as detected by various HPE-CNNs, as well as players’ true motion, as obtained from our manually labeled dataset.

### 1.1. Related Work

The general task of locating one or several objects in a scene over time while keeping track of their identity is commonly known as object tracking or simply tracking. Tracking has applications in a wide range of domains, and therefore has been heavily researched in various communities. For example, in sport sciences, a specific athlete is tracked over time in a preferably unobtrusive manner during training and competition, thereby collecting movement data for measuring performance or game tactical aspects.

Different approaches have been developed for tracking athletes and applied to various sports. Early work dates back to the 1970s, where the movement and work-rates of different positions in English soccer players were analyzed using a manual notation system [16]. Today, more sophisticated approaches are available, due to computer technology improving in terms of computing power, size (miniaturization), energy consumption and affordability.

For example, in their work, Kirkup et al. [17] demonstrate that the localizing of indoor basketball players can be achieved by their wearing a lightweight combination of an acceleration sensor and a radio frequency transmitting beacon. With the development of modern camera technology and its wider and more affordable availability, computer vision has become an important research field for human motion capture [18] in general, and in position localization in particular.

A single camera can potentially be used to obtain the essential movement data of several athletes simultaneously in an effective and contact-free manner. Different approaches such as marker- and non-marker-based methods have been described [19,20]. Additionally, the broad public interest in various sports and the free-of-charge availability of online video-sharing platforms, such as YouTube results, in the availability of a considerable amount of video data, which can be used for retrospective analyses. Apart from the use of markers, computer-vision-based tracking systems can be classified into either multi-camera or single-camera systems, where camera technology can be conventional, e.g., monochrome, color (RGB), or more advanced, such as depth cameras based on structured light or time-of-flight measurements.

While it is not possible to perform a full three-dimensional reconstruction of a single RGB-camera image, due to the lack of depth information, methods using depth-sensitive cameras have been evaluated for human motion tracking [21,22]. In this regard, Microsoft’s Kinect v1 and v2 were of special interest. The first, introduced to the gaming market in 2010, utilizes a structured light depth sensor and the second, introduced in 2014, is based on a time-of-flight depth sensor. In 2013, Choppin and Wheat investigated the Kinect’s potential for biomechanical sports analysis, including player tracking, and concluded that the v1 could potentially be used in coaching and education situations [23]. These findings were confirmed by He et al. for the sport of badminton, where the Kinect v1 is used to guide training, reduce movement intensity and improve the overall training efficiency [24]. For home-based training, van Diest et al. found that the Kinect System was able to accurately identify all relevant body features needed for motion capture [25]. Further hardware improvements led to the introduction of the Kinect v2, featuring new active infrared capabilities, higher resolution camera and new depth sensing. The v2 has been investigated by Alabbasi et al. regarding its potential for human motion tracking. The authors found higher accuracy and concluded that the new version is capable of real-time motion sensing for rehabilitation and physical training exercises [26]. Another application can be found in balance training for the elderly, where a Kinect System is used to characterize a person’s ability to maintain static or dynamic balance [27].

Outside of the Kinect, more general approaches do not rely on depth cameras, and instead use multi-camera systems, which are either marker- or non-marker-based. Here, multiple-view geometry is applied, which uses point correspondences in several views to reconstruct depth information [28]. An overview of different systems for different sports applications is given by van der Kruk and Reijne [3]. All systems are used for a variety of purposes, from entertainment and training to medical applications. As these systems provide valuable data on the athlete’s performance in competition or training situations, they are of particular value for optimizing training and match preparation. Beyond gaining insight into individual performances, motion tracking can also be used to analyze team performances [29].

One technology that has gained attention due to its major impact in the field of machine learning is human pose estimation (HPE). In computer vision, this task is defined as fitting or finding certain keypoints (joints) of a single person or multiple people and connecting them (bones) to form a human skeleton. For multi-person HPE, there are basically two different methods to distinguish between. The top-down approach first tries to detect all people individually in the image, and then estimates individual poses. For a bottom-up approach, keypoints are detected individually, and then subsequently grouped and assigned to individual people. A comprehensive survey of deep-learning-based HPE is given by Chen et al. [30].

In addition, other algorithms from the field of (deep) machine learning are used, besides HPE, for different tasks in various sports. Table 1 shows an overview of the publications, together with the corresponding sports. As can be seen, object and player segmentation or detection has been explored and applied to a variety of sport applications. HPE and classification are the main, but not the only, research topics and applications for machine learning in sports. Moreover, publications are not limited to a single activity, but are broadly distributed across different sports. As an example, Liang et al. [31] proposed a K-Shortest Path (KSP) algorithm to track multiple player detections, obtained by a CNN, together with a re-identification algorithm based on support vector machines (SVM) in basketball. In [32], an object detector is used in conjunction with an HPE-CNN to infer player possession of either a frisbee or a ball. Similarly, for the sport of curling, an object detector was used for player and curling stone detection [33]. Other HPE-CNN applications include speed detection in running [34] and player pose analysis in tennis [35]. Other machine learning approaches have been used for the detection and classification of direction changes in tennis [36] and referee signal recognition in basketball [37]. Other applications use sensor data instead of images. As an example, Anand et al. [38] proposed a system for swing detection and shot classification with a CNN and bidirectional long short-term memory (BLSTM) in racket sports for tennis, badminton and squash, based on wearable motion sensor data.

### 1.2. Organization

Here, we focus on the racket and ball sport of squash, and aim for player localization and motion tracking by detecting players’ feet using HPE-CNNs in video images. We evaluated existing HPE-CNNs first, before considering the creation of a new one, and examined their applicability to the sport of squash. Therefore, our methods section begins with an outline of our general approach. We then introduce our selection criteria and the related selection process necessary for answering RQ1. Then, we introduce our annotation software, which is used to label feet and event information in squash videos to create a labeled dataset. After presenting a video selection of which HPE-CNNs performed inference, our evaluation procedure is introduced, which aggregates HPE-CNN detections and dataset labels. The evaluation procedure is used to answer RQ2. Finally, we present our method for visual comparison of HPE-CNN detections and dataset labels to answer RQ3. Subsequently the results are presented. After the discussion, the paper is concluded, and closes with our future work.

## 2. Materials and Methods

Figure 1 illustrates our method, where numbering corresponds to presented sections. Colors group logical components, necessary for answering the research questions (RQ1–RQ3): Section 2.1 covers our selection process of including freely available HPE-CNNs for inference in our analysis (RQ1). Section 2.2 is dedicated to the creation of test data, together with the correct labels for inference and evaluation. Creating correct labels requires a tool to manually annotate (label) every frame of all videos. The requirements and implementation are covered in Section 2.2.1. Data creation also requires the selection of suitable videos (Section 2.2.2) from freely available squash matches, covering a wide variety of recording scenarios. The labeling process, where all videos were manually annotated, is covered in Section 2.2.3. In our case, labeling the data requires finding the correct x-, y-coordinate of the center of each players’ left and right foot. Section 2.3 is concerned with our procedure for evaluating the performance of detections obtained by Section 2.1 selected HPE-CNNs based on the test data created in Section 2.2 (RQ2). Finally, Section 2.4 presents a visualization technique for displaying spatial distribution of player locations on court (RQ3).

### 2.1. Selecting CNNs

We conduct a structured search based on the following criteria (C${}_{0}$–C${}_{6}$), which we identified to answer RQ1:C_0_:Multi-Person/Multi-Feet detection;C_1_:Published and implemented source code;C_2_:Code must be up to date;C_3_:Available pre-trained model weights;C_4_:Machine-readable output;C_5_:Demo/Showcase application;C_6_:Reasonable effort in setting up.

To find a list of publicly available HPE-CNNs, we searched on paperswithcode [44] (date: 27 April 2020), which is a community-driven project. The community’s goal is to provide a free resource for everyone, covering various machine learning tasks and linking them to their available implementations.

As shown in Figure 2, our initial search for “Pose Estimation Algorithms” results in a total amount of 257 articles, which constitutes the starting point of our selection process. As these contain pose estimation algorithms for full human body, hand or head and animal poses, we narrowed our search to the field of Multi-Person Pose Estimation. Multi-Person in contrast to Single-Person is needed due to the fact that in singles squash two athletes are visible at a time. After applying criteria (C${}_{0}$) and dropping duplicates in the result, 23 articles remained. Investigating the linked source-code repositories to check if the code is published and implemented (C${}_{1}$) resulted in dropping another six algorithms. The reasons are either incomplete implementations or empty repositories where the authors have not yet published their code. Dropping another algorithm for an outdated code base (C${}_{2}$), 16 algorithms are still under consideration. Next, we limit our search to publications which provide a pre-trained neural network and therefore the corresponding model weights (C${}_{3}$). This is needed to skip the time and resource consuming process of training a complex neural network. The reasons for not meeting the criterion are first missing model weights, second dead links and finally a broken download archive. Thus, we considered 11 algorithms and investigated if they provide a machine readable output or offer an export of their estimation in C${}_{4}$. This is necessary to evaluate the algorithms accuracy and compare them in different scenarios. While two algorithms could be extended with minimal modifications, six others could not be extended easily. Reviewing the remaining repositories and looking for a demo/showcase application, where we can input our custom video/image list (C${}_{5}$), we had to drop another algorithm. This algorithm’s showcases only allowed for operation on a predefined training and validation set. Finally, after following the setup instructions for each algorithm, one did not lead to any results (C${}_{6}$), and was therefore excluded.

All three algorithms differ in their processing speed, as shown in Table 2. For A${}_{0}$, the authors state a runtime of 220 ms f^−1^ ( 4.55 f s^−1^) on a single core Intel Xeon 2.70 GHz [45]. According to the authors of A${}_{1}$, the runtime depends on two processing phases. The first one, the person part detection, attains a constant runtime of 99.6 ms f^−1^, which is independent of the number of visible people in the image. The second processing step, which merges the detections, achieves a speed of 0.58 ms f^−1^ for 9 people. In general, the authors report a total runtime of 113.64 ms f^−1^ ( 8.8 f s^−1^) for 19 people on a laptop with an NVIDIA GeForce GTX-1080 GPU [46]. For algorithm A${}_{2}$ we use the TensorFlow.js version, which is implemented in ml5.js. This implementation’s performance is dependent on user-definable variables such as the input image scale factor and the output stride, which affects the internal shape of the network layers. Using the default values provided by the implementation shows a performance of ≈ 100 ms f^−1^ ( 10 fs^−1^) on an off-the-shelf laptop. Every algorithm variant processes all resampled frames for each of the four videos V${}_{0}$–V${}_{3}$.

All three selected HPE-CNNs differ in terms of their architecture. As shown in Table 2, A${}_{0}$ shows the deepest architecture using a ResNet-101 [52] with an adapted stride for the body part detection [45]. A${}_{1}$ is composed of fewer layers by using the first 10 layers of VGG-19 to initially create feature maps, which are then processed in six two-branch stages. The first stage stages’ branches consist of each five convolutional layers, while the remaining five successive stages are composed of seven convolutional layers each. The flattest network is A${}_{2}$, as the 28 layer deep MobileNetV1 is used.

### 2.2. Creating a Labeled Squash Dataset

To evaluate the accuracy of the resulting HPE-CNNs, a domain-specific dataset is required. Datasets containing ground truth information for articulated human body pose estimation [48] or object detection [49] exist. However, as of today, there is no available dataset for squash players on the court. In Section 2.2.1, we start by assembling a list of requirements for a tool to annotate squash video data for feet detection purposes. We then evaluate existing software tools against these requirements and present our own custom-developed, dedicated annotation tool. In Section 2.2.2, we then describe our process for selecting real-world squash videos for inclusion in this study, show their specialties and describe our applied preprocessing procedure. In the last Section 2.2.3, we show our actual annotation procedure, which finally leads to the labeled dataset.

#### 2.2.1. Annotation Tool

To evaluate the accuracy of feet positions detected in video data by modern machine learning algorithms, a dataset with known labels is required. However, as of today, there is no available dataset for squash players on the court. In order to create the necessary dataset, a software tool is needed which fulfills the following requirements (R${}_{0}$–R${}_{4}$):R_0_:Step through single frames in videos;R_1_:Assign identifiers to objects of interest (OOI);R_2_:Annotate OOI locations as points in frames;R_3_:Annotate events for individual frames;R_4_:Export annotations in a machine-readable format.

All requirements are essential in our context of creating spatial labels on players’ feet in every frame in a video, and evaluating them in our toolkit. Requirement R${}_{3}$ is of special interest, as it allows for the annotation of additional non-spatial meta information, i.e., the start and end of a rally. An online search for existing free tools, which are suitable for annotation in machine learning, led to PixelAnnotationTool (PAT) [53], LabelMe (LM) [54] and Computer Vision Annotation Tool (CVAT) [55]. Apart from PAT, these offer the possibility of processing videos on a frame-by-frame basis and thus match requirement R${}_{0}$. For PAT, a preceding extraction of single images would be necessary. In addition, PAT does not completely fulfill requirement R${}_{1}$, since it primarily represents a semantic rather than an instance classification, while both other tools meet this requirement. As we want to annotate feet positions as individual points in frames (R${}_{2}$), a tool must be able to annotate individual points, which is only implemented in CVAT, while PAT and LM are only able to annotate box/polygon shapes. However, one could interpret the barycenter of the box or the polygon as the desired point location. Regarding the machine readable export, all three tools comply with requirement R${}_{4}$. However, none of them can handle the annotation of non-spatial events requirement (R${}_{3}$). A complete comparison is shown in Table 3. As can be seen, neither PAT, LM nor CVAT entirely fulfill our requirements, necessitating the creation of a custom tool.

We developed a tool which is able to perform custom instance labelling of objects using single points combined with an identifier (ID) for each frame in a video individually. By using the same ID over multiple frames, the object is tracked through the entire frame-by-frame sequence. As shown in Figure 3, we additionally implemented eventgroups (EG) in our tool. An EG (e.g., game state) is basically a set of an arbitrary number of customizable events (e.g., non-rally, rally) which may occur in frames. Note that events inside a single EG are disjointed in pairs, as only a single event per EG can occur in a single frame.

When exporting annotated video data, our tool exports two separate files in JavaScript Object Notation (JSON), where the first one contains descriptive data (e.g., frame size) together with the corresponding markers as spatial labels in normalized pixel space. The other contains only the event groups and actual events, while both use the corresponding frame number as key for image assignment. Thus our tool satisfies all identified requirements R${}_{0}$–R${}_{4}$, and will therefore be used for annotating videos.

#### 2.2.2. Selecting Videos

To create the necessary labeled dataset for processing and evaluating HPE-CNN detections in squash matches (RQ2), it is necessary to select video recordings. Since we intentionally do not use the SAGIT/Squash system, requiring a bird’s eye view of the court, we are free to choose from a vast number of online available squash matches. For that purpose, we carried out a search on the most popular video-sharing platform, [56], for different squash matches. We decided to select four different courts with different conditions. Figure 4 shows one example frame for each selected video.

The first video V${}_{0}$ shows a basic squash court, which can typically can found in a racket sports center. It shows the well-known white front and side walls with line markings in red. Only the back wall is made of glass. The court in video V${}_{1}$ is representative of a typical indoor show court, featuring four glass walls. This results in reflections of both players during the match. What happens if a glass court is located outside, e.g., for a big world tour event, is shown in V${}_{2}$. The audience is reflected in the back wall and, additionally, there are photographers above the tin behind the front wall. The last video V${}_{3}$ is similar to V${}_{0}$, as it shows a typical indoor court. However, a white supporting beam is located in the middle of the scene, which leads to players being partially occluded. The back wall is still made of glass. All show courts typically use different color schemes for glass tint, line and floor color, as long as they contrast with any other color [57] (10.13).We considered men’s and women’s matches. Table 4 summarizes the differences between V${}_{0}$–V${}_{3}$ and provides some more technical information about the videos, such as their spatial and time resolutions.

Besides the aforementioned variations, some characteristics are shared by all videos: all of them are filmed with a single camera located behind the court, providing nearly the same perspective as RBG-frames for the entire gameplay. In addition, the camera is mounted in a stationary position at its location and does not pan, tilt or zoom. Thus, the provided images are stable and the camera angles do not change.

All of our selected videos were subsequently preprocessed. As can be seen in Table 4 the sum of frames for all videos is 180,160. Assuming that, in each frame, four feet are visible, and that it takes one second to annotate each foot position with a single click, we arrive at a minimum of 720,640 s ≈200 h, which is not practical. Instead of using all frames, we performed a temporal resampling of the entire videos while preserving their corresponding spatial resolutions to reduce the annotation time. For that purpose, we select one frame for every video second and discard all the others. Additionally, keeping only a single frame for every second increases the inter-frame variability, while reducing the temporal resolution. Furthermore, the original video V${}_{2}$ contains recordings from additional cameras from different angles, which were removed before the resampling process. Finally, a total amount of 4332 frames remains.

#### 2.2.3. Annotating Videos

In this section, we describe our procedure to obtain the dataset. For that, we use our software tool, presented in Section 2.2.1 to label the videos selected in Section 2.2.2. For each input video, the following procedure is applied: First, the resampled, preprocessed video is loaded into our annotation tool. Second, an initial list of labels for OOI annotation is created. The list contains identifiers for both players’ feet, where ID = 0, ID = 1 represent the first player, and ID = 2, ID = 3 the second player’s left and right foot. Next, by stepping through the video frame-by-frame, markers are re-positioned by hand to match the correct feet in pixel space in each frame. Beside the markers, we used our custom event system to distinguish between ball in play and ball in hand. For that, we created an event group called “game_state”, which contains “rally_start” and “rally_end” events. These were then assigned to the closest frames in time whenever a sequence of shots began or ended. After finishing this process, the results were exported and stored to disk. Normalization of marker positions was carried out by dividing every pixel coordinate with the video resolution, so that the frame’s origin is located in the top left corner. These files, containing our labels, are publicly available on GitHub and are ready to serve as the ground truth in the evaluation procedure described in the next section.

### 2.3. Evaluation Procedure

Here, we present our evaluation procedure, which is implemented using Python 3.8 with numpy 1.18.1, pandas 1.0.2, scipy 1.4.1, and opencv-python 4.2.0 libraries. It evaluates the HPE-CNNs’ detections together with the ground truth dataset labels. In Section 2.3.1 we begin by presenting our procedure for classifying a detection as correct or not, and present the associated evaluation metrics computed. Subsequently, Section 2.3.2 was dedicated to grouping options for evaluation.

#### 2.3.1. Evaluation Metrics

For evaluation, it is important to decide whether or not a detection correctly matches a label. We intentionally did not use the percentage of correct parts (PCP) or the percentage of correct keypoints (PCK) for this purpose, because we had not annotated the full human poses in our dataset, only the feet keypoints. Instead, our task was evaluated similar to object detection, where we considered the $L^{2}$-Norm pixel distance of the detection with respect to labels at different thresholds. If a detected marker fulfilled a required threshold, it was considered as true positive (TP), and as a false positive (FP) otherwise. In addition, if no labels were detected, then this was referred to as false negative (FN). True negative values were not calculated. This is due to the fact that a true negative would be a correctly undetected label. Based on frame individual TP, FP, and FN values, we then calculated different evaluation metrics. First, we computed the precision (PPV) as the fraction of those detected among all as positive classified instances. Second, we calculated the recall (TPR) as the fraction of all as correct classified instances, divided by all relevant labels (see Equation (Equation 1). Although both are standalone metrics, it is important and common to place them in a relationship. To address this, we also reported the threat score TS and, more importantly, the F${}_{1}$ score, as the harmonic mean of PPV and TPR (Equation (2)):(1)$$\begin{array}{cccc} \mathrm{PPV} & =\frac{\mathrm{TP}}{\mathrm{TP}\;+\;\mathrm{FP}} & \;\mathrm{TPR} & =\frac{\mathrm{TP}}{\mathrm{TP}+\mathrm{FN}} \end{array}$$(2)$$\begin{array}{cccc} \;\mathrm{TS} & =\frac{\mathrm{TP}}{\mathrm{TP}\;+\;\mathrm{FP}\;+\;\mathrm{FN}} & \;F_{1} & =2\;\cdot \;\frac{\mathrm{PPV}\;\cdot \;\mathrm{TPR}}{\mathrm{PPV}\;+\;\mathrm{TPR}} \end{array}$$

Since we evaluated the selected HPE-CNNs using object detection, average precision (AP) was one of the most common metrics used to determine the accuracy. AP was calculated by utilizing the prediction scores (confidences), which are usually provided for each single detection, together with the computed precision and recall values. As well as these metrics, we computed the spatial information. For this, the normalized and absolute pixel errors to the closest matched labels were calculated for the detections individually, and stored on-site with their TP classifications.

#### 2.3.2. Grouping Options for Evaluation

Our evaluation procedure allows for the grouping/observation of results with respect to different characteristics. By knowing the start- and end-of-shot sequences, by using the game states as our event labels, we could infer whether or not a frame was part of a rally. When we observed only frames that were part of a rally, we referred to this as “frames rally” (FR). When considering only non-rally frames, we used the term “frames non rally” (FNR). To include both types of frame, “frames all” (FA) was used.

Since we labeled left and right feet separately, and the HPE-CNNs also report them separately, we differentiated between types used for detection in our evaluation. The first option is to ignore the players’ feet identifier (i.e., left/right) and match every detection with all unmatched labels per frame. Thus, no distinction between left and right feet is made. We refer to this as “match all” (MA). The other possibility is to consider the players’ feet individually, and distinguish between left and right foot detections. For this, we tried to match every individual left foot detection with all unmatched left foot labels, and right detections with right labels. When we evaluated this, we referred to it as “match individually” (MI). For matching, we used the detection that was most similar to our foot labels. In case of A${}_{1}$F${}^{0}$, this is the body model’s heel detection. For the others, only the ankle position was detected and used.

### 2.4. Heatmap Visualizations

In this section, we outline a method and example application of how the detection results can be utilized (RQ3). If the goal is to implement and evaluate a sport-specific training procedure, the spatial location distribution of players during a match is of special interest. Additionally, considering individual player locations may show their strengths and weaknesses, which may help with coaching during an athlete’s training process. For this, we will show how we use heatmaps as a graphical representation technique for marker (label) locations. Heatmaps are a visualization technique using a color scale, which highlight areas according to the amount of time a player spent at that specific location during matchplay. As can be seen in Figure 5, we created two types of heatmap. One is seen from the camera’s perspective, and is used as an overlay image on top of the corresponding video frame for qualitative analysis. The other represents a virtual bird’s eye view of the court. For this view, we estimated the players’ on-court location from the image pixel. We utilized the presence of a well-known calibration object in every frame: the court with its play lines. Using this, camera calibration can be performed and the resulting camera parameters allow for the estimation of the projection of any image pixel onto the court floor.

#### 2.4.1. Overlay Heatmaps

Overlay heatmaps were computed for A${}_{0}$–A${}_{2}$ separately. Each used a M×N accumulator matrix, where M×N×3 corresponds to the resolution of the color input video. Additionally, a 21×21 normalized gaussian kernel, with a standard deviation σ=5 along both axes, was created. For every HPE-CNN detection, the accumulator was increased by adding the kernel, placed with its center at the detection’s (x,y) image coordinate. In the process, locations with more detections were valued more highly in the accumulator. After adding all detections to the accumulator, the actual heatmap was generated by taking the natural logarithm (after adding one to avoid ln(0)), and normalizing it to the real interval [0,1], respectively [0,255] for one-byte integers. This post-processing was carried out so that very large values did not dominate very small values. The resulting gray scale heatmap can then be colorized with any colormap.

#### 2.4.2. Top-Down Heatmaps

For our top-down heatmaps, we used a 975×640 accumulator, as described in [58]. We selected this shape due to the court’s dimensions (9.75 m ×6.40 m), such that 1 px corresponds to 1 cm. When adding a detection, the corresponding world location on the court’s floor is estimated using the camera’s rotation and translation matrices, obtained from a calibration process using the court dimensions. Subsequently, these are converted to the accumulator’s image coordinates. We use the same gaussian kernel and post-processing as described above.

## 3. Results

This section summarizes our results in three different parts. First, we present basic statistics which generally characterize our labeled dataset. Then, we investigate and present the results of our evaluation procedure with respect to the different available metrics and observations (RQ2). Finally, the results of our heatmap visualization are presented (RQ3). Reviewing all videos in combination with labels and non-spatial events, we can count frames with respect to game state and labels (per frame).

### 3.1. Dataset Statistics

Table 5 shows an overview of the dataset when combining videos with their corresponding labels. The dataset is generally very balanced across the videos, with one exception. In V${}_{2}$, a clearly higher percentage of rally frames (86.3%) is present. This is due to the fact that the source video contained perspectives from different cameras, and was preprocessed by cutting out all unsuitable camera angles. Overall, we report 2347 rally and 1985 non-rally frames, which is 54% and 46% of the complete dataset. By inspecting the labels over all videos, we annotated a total of 16,246 feet, which is 3.75 labels per frame. Table 5 also shows the number of feet detected by each HPE-CNN.

### 3.2. HPE-CNN Evaluation Results

As the evaluation procedure allows for the individual observation of, and reports all metrics for “rally” (FR), “non-rally” (FNR), and “all frames” (FA) separately, robustness against occlusion can be tested. As rallies contain regularly occurring occlusions induced by gameplay, and non-rallies are characterized by little or no movement, we calculated the t-test for the means of two independent samples. Thus, the hypothesis is that there are no differences in metrics between FR, FNR, and FA. We performed the test for all recorded metrics individually, and tested them in pairs. The lowest *p*-value with 0.3 is reported for the PPV metric of V${}_{1}$, when considering all- against only-rally frames. However, the highest value is 0.98, which was reported for the TPR metric of V${}_{1}$ when testing rally against non-rally frames. As all other tested values were within [0.3,0.98], no significance can be assumed. Consequently, the hypothesis cannot be rejected, which indicates that there are no differences between FA, FR and FNR. Thus, we conclude that the investigated HPE-CNNs provide robust feet detection even during phases where player occlusion occurs. For this reason, we will consider rally and non-rally frames by using all frames (FA) in all the following results.

#### 3.2.1. Precision for Both Matching Variants

Precision for matching types In Figure 6, the precision values for all HPE-CNNs A${}_{0}$–A${}_{2}$ and for all videos V${}_{0}$–V${}_{3}$ are shown from top to bottom. On the left side, the matching is “match all” MA, which represents the matching of all feet labels without any side distinction. On the right side, however, the matching type results are reported for “match individual” MI, where a distinction between left and right is made.

It can be seen that precision always increases, together with the threshold for all HPE-CNNs and videos. The reason for this is that, with an increasing threshold, the range of true positive classifications also increases. Comparing every HPE-CNN in both matching variants, the precision is lower for individual matching on the right side. However, A${}_{2}$ shows the lowest precision over all threshold stages in all videos, while A${}_{1}$F${}^{0}$ and A${}_{1}$F${}^{2}$ show the highest end values. From this, it can be concluded that A${}_{2}$ is detecting the wrong locations, while the detections of A${}_{1}$F${}^{0}$ and A${}_{1}$F${}^{2}$ are correct. Additionally, A${}_{1}$F${}^{0}$ rises faster, which indicates more correct detections at lower thresholds. Thus, it can be concluded that A${}_{1}$F${}^{0}$ performs the best correct feet detection at lower thresholds. It is important to note that precision is a metric for the correctness of the detected labels only, not for their completeness.

#### 3.2.2. Recall for Both Matching Variants

In Figure 7, the recall metric is shown for all video/HPE-CNN combinations. Similar to Figure 6, the left side represents “match all”, whereas the right side shows the results for “individual matching”. This is reported for all videos, from top to bottom.

Unlike precision, recall is the metric for completeness. It can be seen that the highest end values are reported for A${}_{1}$F${}^{0}$, which also rises faster at lower thresholds. Thus, it can be concluded that the detections of A${}_{1}$F${}^{0}$ show a high degree of correctness. Moreover, it can be concluded that this correctness is attained even at lower pixel thresholds, and thus a higher accuracy is achieved.

#### 3.2.3. Combination of Precision and Recall

Looking at precision and recall individually provides insights, to a certain extent, into the classification results. A high recall corresponds to the completeness in finding labels, whereas a low recall value corresponds to missing labels. A high precision indicates that the found labels are correct, while a low precision indicates that the detections are false positives. A system which detects many labels correctly would have a high precision and a high recall. Therefore, considering them in combination is of particular interest. When looking at the recall for A${}_{1}$F${}^{2}$ detections in V${}_{2}$, it is particularly notable that the value is clearly below the other HPE-CNNs for both matching types. However, it still achieves a high precision on the same video. This combined effect of high precision and low recall shows that A${}_{1}$F${}^{2}$ is missing detections, but, for the found ones, it has a high fidelity.

The opposite is shown in V${}_{2}$ for A${}_{2}$, which reaches high recall values while staying low in terms of precision. This combination indicates a high completeness in finding labels, but, unfortunately, many of the found labels are not classified as correct. Consequently, looking at V${}_{2}$ at the maximum tolerance threshold of 50 px, A${}_{2}$, with a recall for MA (MI) of 0.82 (0.69), is acceptable for finding feet; however, unfortunately, these are often false positives, as the low precision of 0.49 (0.41) indicates. On the other hand, A${}_{1}$F${}^{2}$ is missing labels, with a low recall of 0.45 (0.43), but reports a very high precision of 0.99 (0.93) for the found labels.

As the best values are reported for A${}_{1}$F${}^{0}$, except for precision on V${}_{1}$ and V${}_{2}$, it can be concluded that A${}_{1}$F${}^{0}$ has a high completeness for correct detection on non-glass courts. However, the low precision for videos showing glass courts indicates that the found labels are detected incorrectly. Following this, the conclusion is that the type of court matters. Player reflections lead to false detections and, therefore, reduce the precision metric.

#### 3.2.4. Balanced Metrics

As mentioned, individual consideration of precision and recall is only possible to a certain extent. To deal with both types, TS and the harmonic mean of precision and recall F${}_{1}$ are used. Both metrics show similar results for individual video/HPE-CNN detection combinations. As Table 6 shows, the values are slightly lower for matching variant MI compared to MA. The lowest values for these metrics were reported by A${}_{2}$ for all videos and both matching variants, with one exception: V${}_{3}$ with MA. However, the best values are shown by A${}_{0}$ and A${}_{1}$F${}^{0}$, apart from V${}_{2}$, where A${}_{1}$F${}^{1}$ is very close in the lead. A${}_{1}$F${}^{2}$ never reaches the highest accuracy, compared to the others, for balanced metrics.

Considering only A${}_{0}$ and A${}_{1}$F${}^{0}$, as they report the highest values, there is a difference in the matching variant. For individual matching, A${}_{0}$ never outperforms A${}_{1}$F${}^{0}$. The highest balanced metric for individual matching always shows A${}_{1}$. Furthermore, A${}_{1}$F${}^{0}$ obtained the best results for individual feet matching (apart from V${}_{1}$). This may be a result of the additional feet training data, which were used for the A${}_{1}$F${}^{0}$ model (see Table 4). As already indicated by the precision and recall results, this led to the conclusion that A${}_{1}$F${}^{0}$ can obtain the most accurate results of the considered HPE-CNNs.

#### 3.2.5. Average Precision

We report the average precisions (AP) results for the selected HPE-CNNs, except for A${}_{0}$. This is due to the lack of necessary prediction scores (confidences) during the detection. We investigated individual matching, where left and right feet are distinguished, as this is more restrictive compared to matching all feet detections with all labels. The results are reported at different threshold levels as AP${}_{px}$. Although all AP values are available in our data repository, Table 7 shows an excerpt of the AP values starting from 25 px.

For all videos, A${}_{1}$F${}^{0}$ first reaches an AP of at least 0.9. The required threshold for this is 30 px, except for V${}_{2}$, where it is only 25 px. The other variants of A${}_{1}$ also achieve an AP of at least 0.9, but at higher thresholds. For all videos and thresholds, A${}_{2}$ never reaches an AP of 0.9 on our dataset, even when considering the less restrictive matching type MA. The highest AP ever reached by A${}_{2}$ is 0.89 on V${}_{2}$, with the less restrictive MA. Therefore, it can be concluded that, in terms of AP, A${}_{1}$F${}^{0}$ performs best, as it was trained on additional foot data.

### 3.3. Heatmap Visualization

Figure 8 shows our heatmap visualization method for all combinations of ground truth (GT), HPE-CNNs and videos. To facilitate a visual comparison, each column represents one of the four input videos V${}_{0}$–V${}_{3}$. The first row presents heatmaps generated from annotated labels, which served as the ground truth during evaluation. The other rows show heatmaps obtained from the HPE-CNNs, and include each single-foot detection for both players. For a better comparison, we colorized the logarithmic transformed gray-scale heatmaps using the perceptually uniform magma colormap. Regarding video V${}_{0}$, all HPE-CNNs show visually convincing detection results, similar to the ground truth heatmap. Although they reach different intensities, high values representing high detection density appear in the same heatmap locations across all HPE-CNNs. In V${}_{1}$, dense spots of false detection can be seen for A${}_{1}$F${}^{0}$ and A${}_{1}$F${}^{1}$ at the front wall’s right corner (i). Furthermore, the glass court seems to lead to false detections of the players’ reflections (ii). The same problem is present in V${}_{2}$, with the mirrored audience (iii). It appears that A${}_{1}$F${}^{2}$ is the most robust HPE-CNN when looking at the players’ reflections in V${}_{1}$, and the mirrored audience in V${}_{2}$. However, the support beam in V${}_{3}$ seems to disrupt the detections for A${}_{1}$F${}^{2}$, as detections only appear below that object (iv). Of all the HPE-CNNs, the heatmaps obtained from A${}_{0}$ detections show the highest visual similarity with respect to the heatmaps obtained from labels.

#### 3.3.1. Heatmap Post-Processing

To address the issue of false detections in reflective glass courts, we show the results after applying a domain-specific post-processing step. As only locations on the court should be included in our heatmaps, each detection is checked with respect to a bounding constraint. For that, we utilize the court’s dimensions as domain-specific knowledge and, therefore, we are able to exclude unwanted detections outside the court. Figure 9 visualizes the process exemplary for a single result, whereas many false detections are present on the court’s glass walls and in the front in (Figure 9A). When applying post-processing, all unwanted detections shown in (Figure 9B) are filtered, which results in the improved visualization seen in (Figure 9C).

Since most false detections are reflections of athletes or the mirrored audience on glass surfaces, it can be concluded that the positive effect of post-processing is the best for glass courts to visually improve heatmap results. As a prerequisite for this, knowledge of the scene, in our case, the court, is needed.

#### 3.3.2. Processing Speed

As we investigated trained HPE-CNN models only, we performed no training process using our dataset. Instead, we used it for evaluation during inference to detect athletes’ feet. For the HPE-CNNs we considered, we can confirm the processing speeds relative to each other, as stated by the authors. Here, A${}_{2}$ showed the fastest inference by using a MobileNet architecture, which has the least depth among the CNNs considered in this work. The computation of heatmaps is an iterative process and can be done in real-time, since the algorithm performs a basic accumulation of values in an allocated memory block. Heatmap formation can be visualized after each frame, but can also be done once at the end, which would reduce computation cost. In summary, the limiting factor for heatmap visualization is not their computation but the inference time needed to obtain detections.

## 4. Discussion

Based on the obtained results, we can answer the research questions:RQ1: We found that three different HPE-CNNs out of five variants are ready to use for out-of-the-box inference on squash data for motion analysis;RQ2: Overall, our evaluation procedure has shown sufficient accuracy for the identified HPE-CNNs on a domain-specific squash dataset;RQ3: Our heatmap visualization technique has been shown to technically be able to present detections or labels for visual assessment.

We have investigated open-source and pre-trained CNNs for human body pose estimation (HPE-CNNs). We found three HPE-CNNs that fit our selection criteria (RQ1), and evaluated a total of five variants on our newly created squash dataset to detect and localize player’s feet positions. Our findings on the game state weakly suggest that the rallies and the short breaks in-between are evenly distributed. As we rely on a standard camera perspective, used to broadcast from behind the court, player occlusions occur frequently in rally situations. In non-rally situations, there are fewer occlusions, due to the fact that both players move towards their respective service boxes. The HPE-CNN performance investigations into differences in rallies and non-rallies revealed that all algorithms are robust against occlusions (RQ2). Heatmap visualization can be used to visually assess the quality of HPE-CNN detections with respect to ground truth labels and, therefore, is technically able to serve as a visual inspection tool for coaches and athletes. (RQ3)

### 4.1. HPE-CNN Evaluation

Comparing different matching types on different thresholds reveals differences regarding our evaluation metrics. When presented with no difference between left and right feet, the results are slightly better. Thus, the performance can be improved when no distinction is necessary in the application at hand. In general, all metrics have a better performance with an increasing detection threshold. This is to be expected, as the consideration radius is increased. Comparing precision and recall for different variants of A${}_{1}$ shows that A${}_{1}$F${}^{2}$ reaches high precision values, while these remain relatively low for recall. Consequently, this shows that this variant has too many detections. This does seem to depend more on the training process and training data, as this variant is the only one of A${}_{1}$ which utilizes the MPII dataset. However, A${}_{2}$ seems to confirm this finding, since it used the same dataset during its training process. When selecting a HPE-CNN for plain-foot detection, without any distinction between left and right, we would suggest choosing A${}_{0}$. This is because the balanced metrics show, in three out of four cases, the best results for A${}_{0}$ when no left/right foot distinction is made. On the other hand, A${}_{1}$ is preferred if a distinction between individual feet is necessary, as it shows the best balanced metric results of all the videos. However, in this scenario, the best results are achieved when using A${}_{1}$F${}^{0}$. The reason for this is probably the additional training data used. This is further confirmed, as A${}_{1}$F${}^{0}$ always outperforms the other algorithms in terms of average precision (RQ2).

Since the HPE-CNNs do not differentiate between reflections, it is actually advantageous for one to recognize them. However, since reflections are unwanted artifacts in our context, they had to be evaluated as false positives. We have shown a simple and effective way to eliminate these unwanted reflections in a real-world application. Thus false detections can reduced by implementing a subsequent, domain-specific filtering of the HPE-CNN detections by a bounding constraint. We demonstrated that filtering out false detections can be performed simply and visually improves the heatmaps and respective player-position visualizations.

### 4.2. Decision Flow Chart for HPE-CNN Selection

There is no simple answer to the question of which HPE-CNN is the best for a given application scenario, as different scenarios have special conditions, and are additionally restricted in terms of the use of hard- or software. Computational complexity may also be a factor to consider. Since all selected HPE-CNNs are based on finding keypoints and constructing poses from those keypoints, there is not much difference in their computational complexity. As stated in [51], computational cost is highly dependent on the CNN feature extraction. This is also reported in [46], where it is stated that CNN processing is the limiting factor. Therefore, we created a flow chart, shown in Figure 10, which can be consulted during the selection process and used for decision support.

At the beginning, the decision has to be made as to whether a distinction between left and right feet is essential for the user application, or whether a pure, side-independent detection is sufficient. If a distinction between feet is not relevant, our results show that A${}_{0}$ is the best choice. The next step is to check hardware availability, i.e., GPU accessibility. If there is no GPU in the user’s setup, a trade-off must be made between performance and accuracy, which leads either to A${}_{0}$ (↓ performance, ↑ accuracy) or A${}_{2}$ (↑ performance, ↓ accuracy). The choice must be made carefully, because when A${}_{0}$ is chosen, left/right distinction is not sufficient. In addition, A${}_{2}$ is the method of choice if a GPU is available and a solution running in a javascript-based application is required. If this is not the case, A${}_{1}$ is the method of choice, whether or not a reflective scene is present. However, in case of a non-reflective setup, A${}_{1}$F${}^{0}$ should be used, whereas A${}_{1}$F${}^{2}$ should be chosen for reflective scenes. Additionally, A${}_{1}$F${}^{1}$, with the presented post-processing, may also be used. A${}_{1}$F${}^{0}$ was trained on the most detailed foot model, which should be considered in any case. Furthermore, when selecting a method, it should be considered that A${}_{2}$ is capable of running in a web browser, with some loss of accuracy.

## 5. Conclusions

The general aim of our work is data-driven video analysis for sports applications, using squash as model sport. To this end, we investigated the usability, accuracy, and applicability of pre-trained, state-of-the-art HPE-CNN models in detecting players’ feet in real-world squash videos. Our contributions are sixfold: First, we present a tool which allows for the annotation of arbitrary event and object instances on image sequences. As well as our specific use case of determining “ball in play” game states, this could be used for other applications, for example, winning or losing a rally, shot types (e.g., straight drive, cross-court drive, drop-shot, boast) or referee decisions (e.g., let, no let, stroke, out-of-court). Moreover, this tool is neither limited to squash nor is it limited to sports applications in general. As it processes videos in general, the number of applications is unlimited. Second, we use this labeling tool, together with the squash videos which are readily available on the internet, to create a squash-specific dataset with manually defined labels for player-feet locations and game-state events. Third, we surveyed 257 CNNs for their suitability for use in squash motion analysis. Fourth, out of those, five HPE-CNN models (RQ1) were applied to real-world squash data, and their detection accuracy was evaluated (RQ2) using the labeled dataset. Fifth, we offer decision-making support for selecting one of the presented HPE-CNNs for a specific scenario. Finally, we implemented and used a heatmap visualization technique to visually compare detections with their corresponding labels (RQ3). By applying a bounding which is constrained during domain-specific post-processing, we reduce possible false detections induced by mirrored athlete appearances on glass courts. Therefore, we conclude that the type of court matters when analyzing recorded squash matches using HPE-CNNs. In addition, this shows that basic traditional post-processing can improve the detection results in visualizations. Our findings support the work of other researchers, who have used CNN technology in a variety of sports, including basketball and tennis. In conclusion, the sport of squash can highly benefit from applications based on general-purpose HPE-CNNs (RQ2). In general, CNN-based HPE technology is capable of transforming the fields of sport sciences, training science and training design. It offers new possibilities for contact-free athlete tracking and motion analysis, and therefore opens up new avenues for data-driven insights into sport applications.

### Future Work

In the future, other sports and sports-related scenarios could be investigated. This could lead to practical applications for training design or quantitative performance assessments. Another exciting area is exploring the feasibility of using this technology for individual training optimization and match preparation. As well as the training aspects, injury prevention and rehabilitation are other important topics. For example, HPE-CNNs could be investigated with regard to their potential for measuring individual movement after (sports) injuries or for replacing classic approaches to collecting motion data in rehabilitation research [59]. Additionally, HPE-CNNs can be investigated for use in smart-home environments [60], where it may be exciting to use heatmap representations as input features for other neural networks. Furthermore, multimodal approaches, as proposed for mobile traffic classification [61], including additional sensors, may be investigated for their higher classification of match play strategies and analysis, as shown in human activity recognition [62].

Since we have shown their technical feasibility, in future work we will apply inference and use our heatmap visualization on individuals, and present the results to trainers and athletes for further insight. We also plan further improvements to our visualization, including quantitative analysis and the ability to derive athletes’ individual metrics by tracking their individual motion data. The results will provide a tool for squash coaches to evaluate the data and monitor athletes’ training progress over time. 

## Figures and Tables

**Figure 1 sensors-21-04550-f001:**
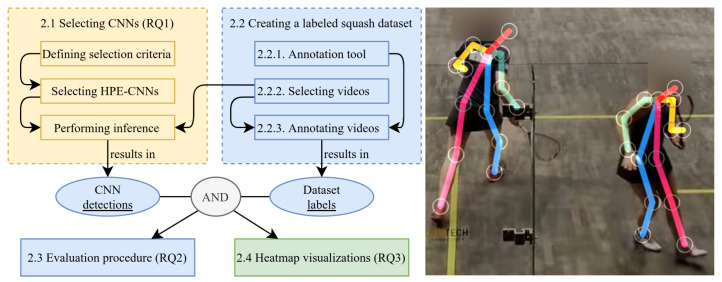
Illustration and example of the different components and their relationship. Left: Section 2.1: Selecting different pre-trained human pose estimation convolutional neural networks (HPE-CNNs) and performing inference, results in CNN detections. Section 2.2: Manually creating a labeled squash dataset by annotating selected videos, results in (ground truth) dataset labels. Section 2.3: Evaluating detections regarding labels using our evaluation procedure. Section 2.4: Utilizing detections and labels for heatmap visualizations. Right: Example detections (circles) of a CNN-based connected human body pose estimator overlaid on the processed video frame.

**Figure 2 sensors-21-04550-f002:**
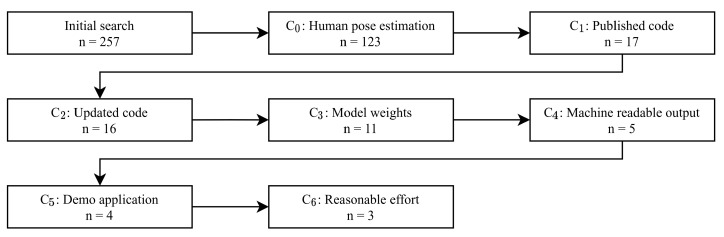
The algorithm selection process. Starting with an initial search, n=257 algorithms are considered. After applying C${}_{0}$–C${}_{6}$, finally n=3 papers remain.

**Figure 3 sensors-21-04550-f003:**
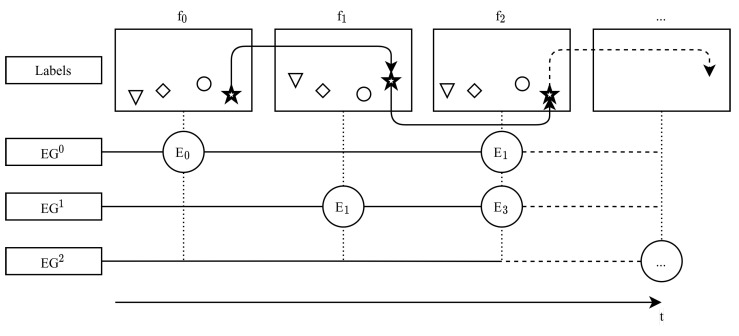
Overview and example of the event group annotation system. Event groups $EG^{0}$ (game state), $EG^{1}$ and $EG^{2}$ are shown as horizontal lines representing time *t*. Frames with spatial markers (feet positions) are shown above and numbered $f_{0}$–$f_{2}$. In $f_{0}$ the event $E_{0}$ (switch to rally) of $EG^{0}$ occurs. Markers are identified in time by a symbol, here shown as different shapes (e.g., star).

**Figure 4 sensors-21-04550-f004:**
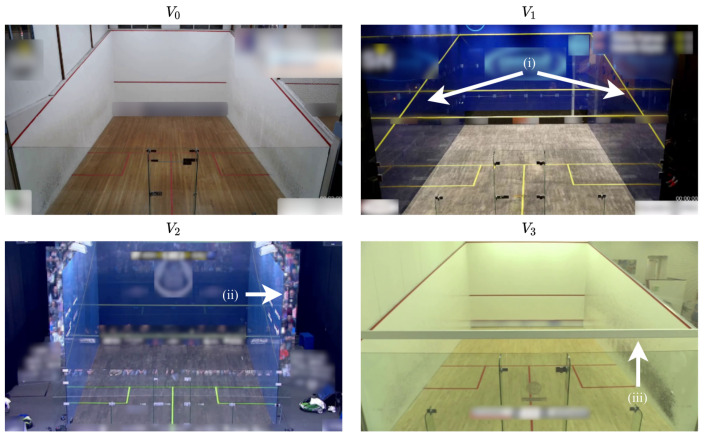
Example frames from videos (**V${}_{0}$**–**V${}_{3}$**), used to evaluate HPE-CNNs. Each image is representative of the special aspect of the respective video. (**V${}_{0}$**) is a standard court situation. (**V${}_{1}$**) shows reflections on the side walls (i) and (**V${}_{2}$**) additionally shows the mirrored audience (ii). In (**V${}_{3}$**), a supporting beam (iii) occludes the image. Please note: due to privacy reasons, players were removed and names, scores and sponsors were blurred.

**Figure 5 sensors-21-04550-f005:**
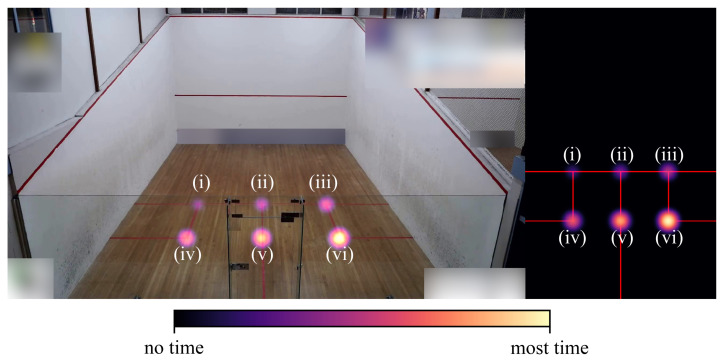
Synthetic heatmaps illustrating the use of a color scale to encode location and time. Heatmap overlaid on a video frame (**left**) and a virtual top-down view (**right**). (i)–(vi) indicate six different court locations and, by means of a color scale, the different durations of time for which a player occupied a location during matchplay. For better visualization, the gaussian kernel was used to increase the accumulator values, and was parameterized with 127×127 px and a standard deviation σ=15 for this figure only.

**Figure 6 sensors-21-04550-f006:**
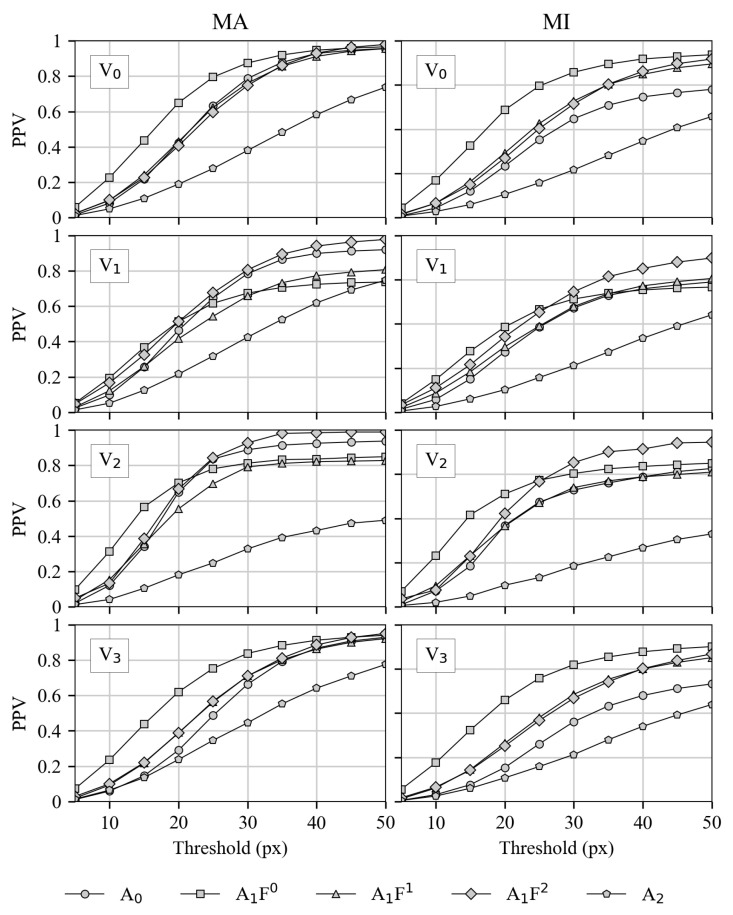
Precisions for all investigated HPE-CNNs and videos at different thresholds with both matching variants. Detections with all labels (MA) are shown on the left, where MI is shown on the right. In rows, results for the four different videos are presented from top (**V${}_{0}$**) to bottom (**V${}_{3}$**).

**Figure 7 sensors-21-04550-f007:**
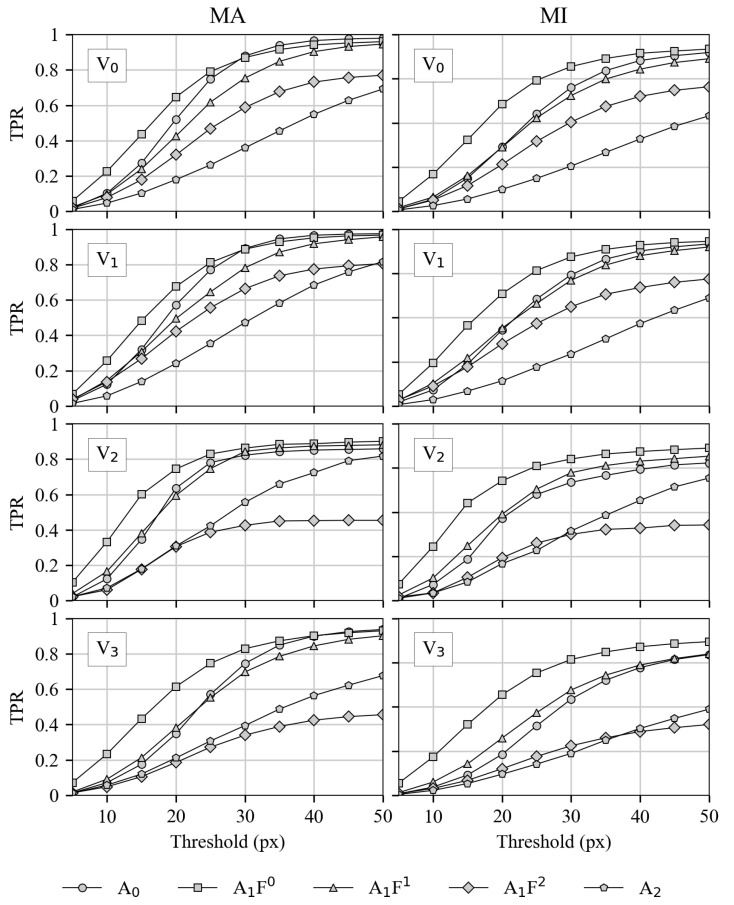
Recall for all investigated HPE-CNNs and videos at different thresholds with both matching variants. Detections with all labels (MA) is shown on the left, whereas MI is shown on the right. In rows, results for the four different videos are presented from top (**V${}_{0}$**) to bottom (**V${}_{3}$**).

**Figure 8 sensors-21-04550-f008:**
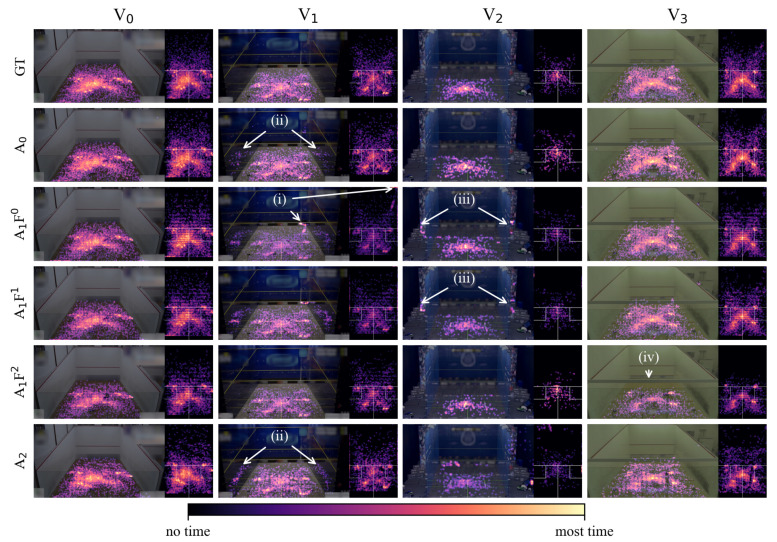
Heatmap visualization reveals differences between HPE-CNN detections and ground truth labels. Each column shows one video V${}_{0}$–V${}_{3}$. The first row depicts the ground truth obtained from labels. Rows 2–6 show detection results for every HPE-CNN evaluated on every video. Heatmaps show labels, and the respective detections for both players. False detections (i)–(iii) result in color differences with respect to the same location in the ground truth heatmap. Reflections of players (ii) and audience (iii) are challenges, specific to glass courts. For A${}_{1}$F${}^{2}$, the support beam in video V${}_{3}$ disrupts detections.

**Figure 9 sensors-21-04550-f009:**
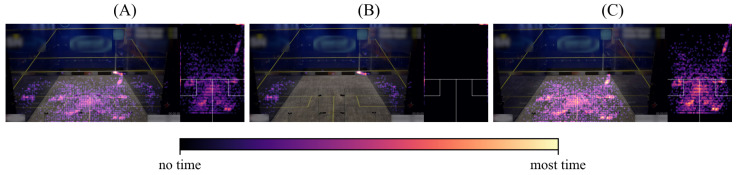
Domain-specific filtering enhances heatmap visualizations by excluding unwanted reflections. In (**A**), the generated heatmap for A${}_{1}$F${}^{0}$ and V${}_{1}$ detections shows reflections and false detections outside the court’s boundaries. Filtering the unwanted detections shown in (**B**) results in a more convincing visualization, as shown in (**C**).

**Figure 10 sensors-21-04550-f010:**
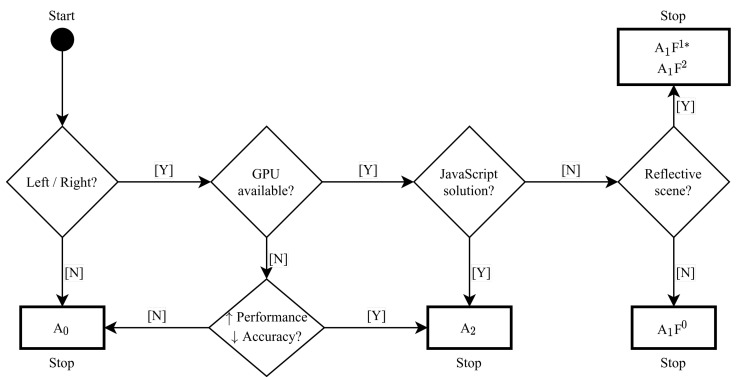
Decision support flow chart for choosing between the five HPE-CNNs. Start with the circle in the top left corner. Diamonds represent decisions and must be answered with [Y]es or [N]o (${}^{*}$, with implemented post-processing).

**Table 1 sensors-21-04550-t001:** Related work regarding applications of machine learning in sports.

Year	Title and Reference	Application
2020	Multi-Player Tracking for Multi-View Sports Videos with Improved K-Shortest Path Algorithm [31]	Basketball
2020	Real-Time Possessing Relationship Detection for Sports Analytics [32]	Frisbee & Football (soccer)
2020	Study on Sports Volleyball Tracking Technology Based on Image Processing and 3D Space Matching [39]	Volleyball
2020	Detection of Ice Hockey Players and Teams via a Two-Phase Cascaded CNN Model [40]	Ice Hockey
2020	Utilizing Mask R-CNN for Waterline Detection in Canoe Sprint Video Analysis [41]	Canoe
2020	FISHnet: Learning to Segment the Silhouettes of Swimmers [42]	Swimming
2020	Human Pose Estimation based Speed Detection System for Running on Treadmill [34]	Running
2019	Analyzing Basketball Movements and Pass Relationships Using Realtime Object Tracking Techniques Based on Deep Learning [29]	Basketball
2019	A machine learning approach for automatic detection and classification of changes of direction from player tracking data in professional tennis [36]	Tennis
2019	YOLO based Intelligent Tracking System for Curling Sport [33]	Curling
2018	Recognition of basketball referee signals from videos using Histogram of Oriented Gradients (HOG) and Support Vector Machine (SVM) [37]	Basketball
2018	Player Pose Analysis in Tennis Video based on Pose Estimation [35]	Tennis
2018	Mask R-CNN and Optical Flow Based Method for Detection and Marking of Handball Actions [14]	Handball
2017	Wearable Motion Sensor Based Analysis of Swing Sports [38]	Tennis, Badminton, Squash
2012	Recognizing tactic patterns in broadcast basketball video using player trajectory [43]	Basketball

**Table 2 sensors-21-04550-t002:** Resulting three algorithms. A${}_{1}$ comes in three different flavors $F^{0}$–$F^{2}$, such that a total of five algorithms can be compared to each other.

Identifier	Name	Training Data	Architecture	Runtime (FPS)
A${}_{0}$	Arttrack [45,47]	MPII [48]	ResNet-101	4.55
A${}_{1}$F${}^{0}$	OpenPose [46]	COCO [49] + Foot [46]	VGG-19	8.8
A${}_{1}$F${}^{1}$	OpenPose [46]	COCO [49]	VGG-19	8.8
A${}_{1}$F${}^{2}$	OpenPose [46]	MPII [48]	VGG-19	8.8
A${}_{2}$	PoseNet [50,51]	COCO [49]	MobileNetV1	10.0

**Table 3 sensors-21-04550-t003:** Compliance table of requirements R${}_{0}$–R${}_{4}$ for different annotation tools. A tool either fulfills the requirement completely (**+**), partially (**/**) or not at all (**–**).

Tool	R${}_{0}$	R${}_{1}$	R${}_{2}$	R${}_{3}$	R${}_{4}$
LM	**+**	**+**	**/**	**–**	**+**
PAT	**–**	**/**	**/**	**–**	**+**
CVAT	**+**	**+**	**+**	**–**	**+**
Our Tool	**+**	**+**	**+**	**+**	**+**

**Table 4 sensors-21-04550-t004:** Additional information for videos included in the dataset.

	Resolution (w,h)	FPS	Frames	Frames Resampled	Court Aspects
V${}_{0}$	(1920, 1080)	50	94,285	1886	default
V${}_{1}$	(1920, 1080)	50	43,435	869	reflective, glass
V${}_{2}$	(1280, 720)	25	3646	146	reflective, glass, mirrored audience
V${}_{3}$	(1920, 1080)	25	38,794	1431	default, white support beam

**Table 5 sensors-21-04550-t005:** Dataset label and detection statistics. Top: Dataset statistics with respect to game state. Bottom: Detections per HPE-CNN for all videos.

		V${}_{0}$		V${}_{1}$		V${}_{2}$		V${}_{3}$	
Rally Frames	(%)	1030	(54.6)	385	(44.3)	126	(86.3)	806	(56.3)
Non-Rally Frames	(%)	856	(45.4)	484	(55.7)	20	(13.7)	625	(43.7)
Total Frames		1886		869		146		1431	
Labels	(per frame)	7253	(3.85)	3346	(3.85)	572	(3.92)	5075	(3.55)
Detections by A${}_{0}$	(%)	9078	(125.1)	4207	(125.7)	575	(100.5)	6114	(120.5)
Detections by A${}_{1}$F${}^{0}$	(%)	7225	(99.6)	4404	(131.6)	603	(105.4)	5032	(99.2)
Detections by A${}_{1}$F${}^{1}$	(%)	7203	(99.3)	3979	(118.9)	609	(106.5)	4985	(98.2)
Detections by A${}_{1}$F${}^{2}$	(%)	5703	(78.6)	2757	(82.4)	261	(45.6)	2434	(48.0)
Detections by A${}_{2}$	(%)	6875	(94.8)	3728	(111.4)	969	(169.4)	4523	(89.1)

**Table 6 sensors-21-04550-t006:** F1 and TS metric results for all video/HPE-CNN combinations at the maximum tolerance threshold of 50  px. Bold: highest Italic: lowest column values.

		V${}_{0}$		V${}_{1}$		V${}_{2}$		V${}_{3}$	
	Metric	MA	MI	MA	MI	MA	MI	MA	MI
A${}_{0}$	F${}_{1}$	**0.968**	0.803	**0.946**	0.817	**0.896**	0.781	0.933	0.724
	TS	**0.937**	0.670	**0.897**	0.691	**0.812**	0.640	0.874	0.568
A${}_{1}$F${}^{0}$	F${}_{1}$	0.963	**0.919**	0.836	0.804	0.874	**0.837**	**0.935**	**0.872**
	TS	0.928	**0.851**	0.718	0.672	0.778	**0.720**	**0.878**	**0.774**
A${}_{1}$F${}^{1}$	F${}_{1}$	0.951	0.866	0.875	**0.822**	0.854	0.788	0.911	0.806
	TS	0.906	0.764	0.778	**0.697**	0.744	0.651	0.837	0.675
A${}_{1}$F${}^{2}$	F${}_{1}$	0.862	0.789	0.883	0.788	0.622	0.586	*0.616*	0.541
	TS	0.757	0.652	0.790	0.650	0.452	0.415	*0.445*	0.371
A${}_{2}$	F${}_{1}$	*0.714*	*0.556*	*0.780*	*0.580*	*0.613*	*0.517*	0.723	*0.516*
	TS	*0.555*	*0.385*	*0.639*	*0.408*	*0.442*	*0.349*	0.566	*0.348*

**Table 7 sensors-21-04550-t007:** Average precisions at different thresholds for MI. Bold: First algorithm’s AP ≥0.9.

		AP${}_{25}$	AP${}_{30}$	AP${}_{35}$	AP${}_{40}$	AP${}_{45}$	AP${}_{50}$
V${}_{0}$	A${}_{1}$F${}^{0}$	0.849	**0.916**	0.947	0.968	0.975	0.979
	A${}_{1}$F${}^{1}$	0.610	0.771	0.871	**0.923**	0.951	0.962
	A${}_{1}$F${}^{2}$	0.518	0.679	0.802	0.878	**0.913**	0.932
	A${}_{2}$	0.240	0.329	0.422	0.509	0.589	0.652
V${}_{1}$	A${}_{1}$F${}^{0}$	0.858	**0.925**	0.952	0.969	0.975	0.978
	A${}_{1}$F${}^{1}$	0.638	0.777	0.866	**0.910**	0.935	0.950
	A${}_{1}$F${}^{2}$	0.643	0.749	0.836	0.874	**0.902**	0.914
	A${}_{2}$	0.224	0.299	0.387	0.473	0.552	0.638
V${}_{2}$	A${}_{1}$F${}^{0}$	**0.936**	0.948	0.963	0.966	0.967	0.969
	A${}_{1}$F${}^{1}$	0.812	**0.907**	0.936	0.951	0.954	0.957
	A${}_{1}$F${}^{2}$	0.819	**0.906**	0.938	0.945	0.952	0.952
	A${}_{2}$	0.454	0.542	0.645	0.720	0.777	0.821
V${}_{3}$	A${}_{1}$F${}^{0}$	0.857	**0.917**	0.947	0.964	0.972	0.976
	A${}_{1}$F${}^{1}$	0.577	0.732	0.823	0.877	**0.905**	0.921
	A${}_{1}$F${}^{2}$	0.524	0.674	0.775	0.847	0.888	**0.911**
	A${}_{2}$	0.223	0.310	0.415	0.502	0.580	0.646

## Data Availability

Our complete tool chain, ranging from the annotation tool (https://github.com/sudochris/pannotator) including the dataset annotations (https://github.com/sudochris/squashfeetdata) up to the evaluation procedure (https://github.com/sudochris/squashfeettoolkit) is freely available at https://github.com/sudochris/squashevaluation (accessed on 30 June 2021).

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
