# Peer review of "Evaluation of Open-Source and Pre-Trained Deep Convolutional Neural Networks Suitable for Player Detection and Motion Analysis in Squash"

_sensors, 2021, doi:10.3390/s21134550_

Round 1

Reviewer 1 Report

Dear Authors,

Thank you for the opportunity to review your manuscript. Below you will find my suggestions to improve the manuscript’s quality.

L24-42: This first paragraph is full of statements regarding sport, but I can’t see any reference supporting such statements. Please, add references in that part.

L36: Just in the first paragraph, the authors are claiming the aims of the study “Here, we are interested in exploring the potential of modern computer vision and machine”. This is not how an introduction should be arranged. Please, make ideas follow more smoothly and end by saying what is the aim and what you have done to achieve it.

L64: Are you referring to a Figure of the manuscript in the introduction? Please amend.

L23-125: This part of the introduction should be written again from scratch. As I have commented, it does not follow a logical flow of ideas. The authors are going and coming again and again without a logical sequence. Please, follow this scheme: what is done, what is missing, what the authors are doing to fill the gap.

L149: Why Youtube is tagged as copyright and Kinect as trade market? What are the differences to tag them differently?

L213: And, after all the (nice) information provided in the introduction, although not properly distributed, the reader does not know what the aims were and what the manuscript is contributing to the field of knowledge described earlier.

L223: Are you proposing a neural network? I am a bit confused because I can barely follow your intentions.

Although I acknowledge the immense work done by the authors, the writing is poor. I am not referring to English language, which is fine, but to other flaws:

  • The text is not presented in logical order. The reader must go up and down to try to understand the meaning.
  • The authors introduced many concepts, variables, measures or procedures without proper explanation, so the reader is lost.
  • In many parts of the text, the reader can’t know if authors are talking about the procedure, software, etc

In the following, I will give general ideas that must be taken into account, together with others that the authors must observe.

L214: Are you talking about a procedure or data collection?

L237: Is this a general scheme?

L244: Is this part of the general scheme?

L272: Did you compare your proposal with other methods?

L288: Is this part of the general scheme?

L353: Is this some sort of review? You made a search with keywords and you resulted in 257 articles, from which you retrieved pose estimation algorithms.

L381: This figure looks definitely like a review

L434: Is this a measure you will use to assess the quality of your system?

L469: Is this part of the system, an example of a necessary step in the procedure?

L485: What is sigma?

L515: What is mu?

L604: Are you introducing a new variable in the results section, not shown in the procedure?

L612: Why are you using three decimal places in Table 7?

My recommendation for the authors is to let colleagues not involved in the study to read the manuscript and ask them what they have understood. The authors will then notice what I am trying to say.

Reviewer 2 Report

The paper is very strong. Although I'm not coming from the field of sports science, I find the research interesting and applicable. I have few suggestions:

  • Line 281, 282, 283 should be one sentece.
  • Lines in Fig.3 should be more visible, especially in V2

For the evaluation metrics, you should definitely consider confusion matrix, and other useful performance metrics from scikit (since you are using it). I think it would benefit the overall presentation of the work.

Reviewer 3 Report

Overall, this is a very nice paper on detection and motion analysis via CNN-based image processing with application to squash. Still, before publication, I would like the authors to address the following comments:

1)    Abstract – “Today, contact-free camera based multi-athlete detection and tracking has become reality mainly..” -> “Today, contact-free camera based multi-athlete detection and tracking have become reality mainly..”

2)    When introducing the benefits of DL, I would like the authors to recall its successful application to manifold verticals:
O’shea, Timothy, and Jakob Hoydis. "An introduction to deep learning for the physical layer." IEEE Transactions on Cognitive Communications and Networking 3.4 (2017): 563-575.

"Mobile encrypted traffic classification using deep learning: Experimental evaluation, lessons learned, and challenges." IEEE Transactions on Network and Service Management 16.2 (2019): 445-458.

3)    Please add an organization paragraph at the end of Sec. 1.

4)    Sec. 2 – I would like the authors to modify rephrase the following paragraph: 

“Neural networks are a special class of machine learning techniques that are fundamentally based on learning by example. During an initial learning phase the neural network is trained to perform a certain recognition, respectively, classification task.
Training a neural network is done by presenting a large amount of example data together with the correct label, representing the answer to the task at hand. Out-of the labeled data a generalization from example to label is constructed using a method called backpropagation. During the following inference phase, new test samples not included in the training data is presented to the network and the corresponding label is predicted (inferred).

as

“Neural networks are a special class of machine learning techniques that are fundamentally based on learning by example. During an initial learning phase the neural network is trained to perform a certain inference task, e.g. classification.
Training a neural network is done by presenting a large amount of example data together with the correct label, representing the answer to the task at hand. Out-of the labeled data a generalization from example to label is constructed using a method called backpropagation. During the following inference phase, new test samples not included in the training data is presented to the network and the corresponding label is predicted (inferred).

Also what do the authors mean by “Out-of the labeled data a generalization from example to label is constructed using a method called backpropagation.”? Please clarify.

5)    It is not clear to me whether the collected and annotated dataset is (will be) made publicly available by the authors. In the latter case, this would greatly foster reproducibility and further advances on the topic.

6)    I would like the authors to briefly discuss the computational complexity involved in the considered HPE-CNNs.

7)    I have appreciated the section on HPE-CNN architecture selection, in which the authors carefully and minutely explain the CNN selection process. According to my understanding, three CNNs are selected after this process. Still, for completeness, it would be very useful providing some details on the CNN architecture associated to these three “survived” methods.

8)    In future work, the authors may want to mention the application of the proposed methodology to detection in unhealthy subjects, e.g.: 

"Effect of Global Postural Rehabilitation program on spatiotemporal gait parameters of parkinsonian patients: a three-dimensional motion analysis study." Neurological Sciences 33.6 (2012): 1337-1343.

And, also, the use of multiple cameras (or alternative sensors), to perform CNN-based multimodal approaches, following e.g.:

"MIMETIC: Mobile encrypted traffic classification using multimodal deep learning." Computer Networks 165 (2019): 106944.

Round 2

Reviewer 1 Report

Dear Editors,

The authors have conducted all the changes required, so I am now happy with the manuscript.

Thank you.

Reviewer 3 Report

Overall, this is a very nice paper on detection and motion analysis via CNN-based image processing with application to squash. 

The authors have satisfactorily addressed my previous comments and modified their manuscript accordingly. Hence, I am glad to recommend the present work for publication.